# The Extracellular Matrix Environment of Clear Cell Renal Cell Carcinoma

**DOI:** 10.3390/cancers14174072

**Published:** 2022-08-23

**Authors:** Leif Oxburgh

**Affiliations:** Rogosin Institute, 310 East 67th St., Room 2-43, New York, NY 10065, USA; leo9022@nyp.org

**Keywords:** collagen, fibronectin, TGFBI, tenascin, periostin, kidney cortex, basement membrane, interstitial matrix, cancer-associated fibroblasts, CAFs

## Abstract

**Simple Summary:**

The extracellular matrix (ECM) controls fundamental properties of tumors, including growth, blood vessel investment, and invasion. The ECM defines rigidity of tumor tissue and individual ECM proteins have distinct biological effects on tumor cells. This article reviews the composition and biological functions of the ECM of clear cell renal cell carcinoma (ccRCC). The most frequent initiating genetic mutation in ccRCC inactivates the *VHL* gene, which plays a direct role in organizing the ECM. This is predicted to result in local ECM modification, which promotes the growth of tumor cells and the invasion of blood vessels. Later in tumor growth, connective tissue cells are recruited, which are predicted to produce large amounts of ECM, affecting the growth and invasive behaviors of tumor cells. Strategies to therapeutically control the ECM are under active investigation and a better understanding of the ccRCC ECM will determine the applicability of ECM-modifying drugs to this type of cancer.

**Abstract:**

The extracellular matrix (ECM) of tumors is a complex mix of components characteristic of the tissue of origin. In the majority of clear cell renal cell carcinomas (ccRCCs), the tumor suppressor VHL is inactivated. VHL controls matrix organization and its loss promotes a loosely organized and angiogenic matrix, predicted to be an early step in tumor formation. During tumor evolution, cancer-associated fibroblasts (CAFs) accumulate, and they are predicted to produce abundant ECM. The ccRCC ECM composition qualitatively resembles that of the healthy kidney cortex in which the tumor arises, but there are important differences. One is the quantitative difference between a healthy cortex ECM and a tumor ECM; a tumor ECM contains a higher proportion of interstitial matrix components and a lower proportion of basement membrane components. Another is the breakdown of tissue compartments in the tumor with mixing of ECM components that are physically separated in healthy kidney cortex. Numerous studies reviewed in this work reveal effects of specific ECM components on the growth and invasive behaviors of ccRCCs, and extrapolation from other work suggests an important role for ECM in controlling ccRCC tumor rigidity, which is predicted to be a key determinant of invasive behavior.

## 1. Introduction

Several different primary tumors have been identified in the kidney; renal cell carcinoma (RCC) is the most common and one of the 10 most prevalent cancers overall [1]. In clear cell renal cell carcinoma (ccRCC), which is the most common subtype of RCC, clusters of tumor cells with clear cytoplasm are found in the kidney cortex [2]. The tumor suppressor *VHL* is lost in over 90% of ccRCCs through genetic or epigenetic mechanisms [3]. *VHL* plays a central role in cellular oxygen sensing [4,5], and its inactivation causes persistent pseudo-hypoxia, resulting in a strong angiogenic profile of tumors [6]. Evidence suggests that the cells of origin for ccRCC are epithelial cells that line the proximal tubule [7,8], one of the most metabolically active cell types in the human body [9]. Transformed epithelial cells in the tumor are surrounded by a network of stroma containing vasculature, interstitial fibroblasts and inflammatory and immune cells [10]. The cellular tumor components are embedded in the extracellular matrix (ECM).

Many fundamental properties of tumors are controlled by their ECM environment [11], including proliferation [12,13], vascularization [14,15] and invasion [16], making the ECM a key determinant of malignancy. Interplay between the ECM and tumor cells is complex and multifactorial. ECM provides the substrate for cancer cell attachment, and determines rigidity, which has a strong influence on malignancy [17]. It also controls signal transduction in cancer cells, exerting a strong influence on their behaviors [18].

The ECM is a mix of components with distinct physicochemical and signaling properties [19,20]. For the reasons outlined above, the ECM composition characteristic of a tumor type is predicted to play a central role in determining the behavior of that tumor. The ECM profile of a tumor is likely characteristic of the organ from which it arises, but also characteristic of the tumor’s cellular composition, with cancer-associated fibroblasts (CAFs) playing a major role in producing the tumor ECM [21]. Difficulties in isolating and maintaining CAFs have hampered experimental studies of these cells in ccRCC. The finding that the outgrowth of CAFs may be dependent on culture conditions that provide the complex ECM environment that they, themselves, generate highlights one important obstacle in the characterization of CAF biology [22]. Indeed, in modeling ccRCC in vitro, we have demonstrated that the CAF population can be maintained by culturing it in an ECM environment that mimics the tumor of origin [23]. CAFs have features in common with activated myofibroblasts seen in fibrosis [24], and one interesting possibility is that insults that convert growth-suppressed fibroblasts in a healthy kidney to a proliferative and synthetic myofibroblast state may provide the appropriate environment for tumor formation from proximal tubule epithelial cells in which *VHL* has been mutated.

The Cancer Genome Atlas includes RNA-seq data for over 500 ccRCC tumors, along with information on patient outcomes, providing a popular resource for studies of association between gene expression and survival that may have prognostic value [25]. Using this resource, a gene signature has been identified that is comprised of 12 genes functionally associated with the ECM [26]. This dataset has also been used to derive a proxy signature for CAF investment in ccRCC, which anti-correlates with survival, suggesting that CAF infiltration in tumors is associated with poor prognosis [27]. While these correlative in silico studies cannot establish causality, they do suggest important relationships between ECM deposition, CAF abundance and tumor aggressiveness. This review discusses published work on the ECM composition of ccRCC, the cellular sources of the matrix components, and the effects of these proteins on cells within tumors.

## 2. VHL Regulation of the ECM

VHL plays a central role in cellular oxygen sensing by ubiquitinating the HIFα transcription factors in normoxia and targeting them for destruction. In the normoxic state, prolyl hydroxylase domain (PHD) proteins hydroxylate proline residues at the amino terminus of the HIFαs, allowing VHL to bind [28]. The function of the PHDs is strictly oxygen-dependent, and when oxygen becomes scarce, PHD hydroxylation is reduced, the VHL-dependent degradation of HIFαs ceases and they accumulate, dimerizing with HIFβs and localizing to the nucleus where they control a hypoxia-dependent gene expression program that governs cellular metabolism, angiogenic signaling and other pathways [29,30].

In addition to this essential hypoxia-sensing function involving HIFαs, VHL controls ECM organization. This has emerged as an important factor in the interaction of *VHL*-inactivated cells with their extracellular environment and is predicted to regulate angiogenesis, invasiveness and metastasis. Studies of VHL disease variants have shown that variants associated with RCC fail to bind fibronectin, in contrast to wild-type VHL [31,32]. The interaction of VHL with fibronectin is dependent on neddylation, in which NEDD8 is ligated to VHL in a process similar to ubiquitination [33]. This covalent modification acts as a molecular switch, converting a fraction of the intracellular VHL pool to a fibronectin-binding form that does not participate in the degradation of HIFαs [34]. Cultured cells were used to demonstrate the physical interaction of VHL with fibronectin in the cytoplasm, and gain-of-function experiments with 786-O ccRCC cells that lack the expression of a functional VHL revealed that VHL is required for extracellular deposition of fibronectin matrix in its fibrillar form [35]. The fibrillogenesis of fibronectin is a force-dependent process in which extracellular fibronectin is bound to integrin receptors α5β1 [36,37] or αVβ3 [38]. Fibronectin-induced clustering of integrins at the cell surface promotes the interactions of focal adhesion kinase and its partners, which initiate actin polymerization and intracellular signaling [39,40]. The polymerization of actin and the modification of the cytoskeletal network applies traction force to the fibronectin molecules that are anchored between integrins or between integrin and another binding partner [41,42]. Stretching of fibronectin unfolds the molecule and exposes cryptic epitopes, greatly increasing the binding of additional soluble fibronectin, and inducing fibrillar matrix formation [43].

The finding that VHL is required for fibronectin matrix formation is supported by studies of *Vhl* mutant mouse embryos, which reveal an absence of fibronectin matrix deposition [35]. The physical interaction between VHL and fibronectin, coupled with the diminutive fibronectin deposition by VHL mutant tumor cells, suggest that the cellular phenotype may be explained by an intracellular build-up of unprocessed fibronectin. However, despite the fact that the transgenic expression of VHL reverses the phenotype and promotes the deposition of fibronectin fibrils, no significant difference in intracellular fibronectin content was observed between 786-O cells with or without transgenic VHL expression [44]. This observation suggests that additional molecular players may be involved in VHL-mediated fibronectin assembly and organization. The fibrillogenesis of fibronectin is an integrin-dependent process [41], and studies of 786-O cells using integrin β1-activating and -blocking antibodies revealed that fibronectin organization into fibrils could be promoted in these cells by activating integrin β1 outside-in signaling [44]. This finding suggests that VHL is required for integrin β1 function, which in turn, is required for fibronectin fibrillogenesis. While fibronectin is generally considered an interstitial matrix (IM) component, its fibrillar form is associated with the BM, and may be important for structural integrity [45]. The loss of fibronectin fibrillogenesis as an early event in the transformation of proximal tubule cells is supported by several reports; however, protein expression analysis using validated antisera does not unambiguously show fibronectin expression in the extracellular space surrounding proximal tubules in adult human kidney biopsies (www.proteinatlas.org, accessed on 12 August 2022), introducing a significant caveat to this aspect of the model for local basement membrane (BM) degeneration following the loss of VHL. Whether or not fibronectin fibrillogenesis is a major contributor to changes in the interaction between the proximal tubule epithelial cell and its BM when VHL is inactivated, changes in integrin associations with other matrix components are likely to be affected since VHL has been shown to be physically associated with integrin β1 and to control its half-life [44,46].

Collagen IVα2 is a VHL interaction partner in 786-O ccRCC cells that express wild-type VHL, but not in cells that express disease variants of VHL. Collagen IVα2 and VHL colocalize at the endoplasmic reticulum membrane, and this association is required for the assembly of the triple-helical collagen IV matrix [47]. Xenografting studies reveal that 786-O tumor cells expressing wild-type VHL form tumors with a tightly packed collagen matrix, whereas 786-O cells either without functional VHL or expressing disease variants form tumors with a loose collagen matrix and more extensive blood vessel infiltration [48]. Interestingly, the angiogenic tumor phenotype is seen in cells expressing VHL variants in which the HIFα-degrading function remains intact but the collagen IVα2-interacting function is lost [47,48]. A comparison was made between tumors derived from 786-O cells expressing VHL with inactivated HIF1α degradation and tumors derived from 786-O cells expressing VHL with intact HIF1α degradation but inactivated collagen/fibronectin interaction; it indicated that the elevation of the angiogenic growth factor VEGF by stabilized HIF1α is not sufficient to generate the highly vascular tumor phenotype. Rather, the disruption of ECM organization is required to promote the highly angiogenic tumor phenotype [48]. Ubiquitome analysis has identified additional candidate ECM targets of VHL such as TGFBI [49], which may influence the invasiveness of ccRCC cells [50].

Based on these studies, one model for the initial events in the formation of a ccRCC tumor (Figure 1) would be that VHL plays an essential role in organizing the matrix immediately surrounding the proximal tubule epithelial cell prior to the somatic mutation of *VHL* and the loss of its matrix-organizing function. The *VHL* mutant cell would alter its turn-over of integrins and relax or perhaps even dissolve its BM locally, generating a loose matrix microenvironment conducive to vascularization while engaging its HIFα proangiogenic program. Studies of ccRCC cell lines with or without the expression of wild-type VHL indicate that VHL suppresses the expression of matrix metallopeptidase (MMP) 2 and 9, while amplifying the expression of tissue inhibitor of metalloproteinase (TIMP) 1 and 2 [51], and that the level of MMP2 expression in ccRCC cells correlates with their capacity for invasion in three-dimensional cell culture [48]. Thus, in addition to local ECM modification with the formation of an angiogenic niche, the loss of VHL would promote the expression of MMP, which would cause ECM degradation and invasive behavior of the tumor cells. This model of VHL-dependent ECM changes that initiate tumor formation is derived from cell culture and xenografting studies, and how it relates to the ECM environment in patient ccRCC tumors at the time of diagnosis is not clear. Based on genetic studies of patient material, the genetic inactivation of *VHL* often precedes the diagnosis of ccRCC by decades, and tumor formation is predicted to be an indolent process [52]. Furthermore, the mutation of *VHL* alone does not appear to be sufficient to initiate tumor formation, as patients with loss of both copies of *VHL* can show benign lesions including cysts [53,54]. Whether local ECM modification caused by the loss of *VHL* predisposes cells to genetic events that lead benign lesions to become malignant is not known, but structural changes associated with cyst formation do suggest that matrix remodeling already occurs in the benign lesion.

## 3. ECM Composition of ccRCC Tumors

Clinical material presents a complex matrix profile compared with cell line and short-term xenografting studies, since the tumor at the time of resection is perhaps decades old and composed of many different cell types. While the information that this type of material can yield regarding initiating events in the formation of the tumor is very limited, it does provide an important snapshot of the matrix environment at the time of surgery, and can potentially be correlated with invasiveness and metastasis. Additionally, recreating the matrix environment of the tumor is an important foundation for culturing primary tumor cells. We investigated the repertoire of ECM molecules in patient-derived ccRCC tumor samples with the goal of defining matrix components that could be defined as characteristic of ccRCC [23]. Seven stage pT3 tumors were collected from consenting patients undergoing partial nephrectomy along with healthy neighboring kidney cortex tissue. We designate the neighboring tissue “healthy” because it has a tissue layout and cell types characteristic of a healthy kidney cortex. Whether the cortex tissue neighboring the tumor is genuinely healthy or whether it is affected by proximity to the tumor is not known, and this is a general feature of studies that compare resected tumor tissue with healthy margins. The tumors and matched healthy cortices were analyzed via mass spectrometry using sequential window acquisition of all theoretical fragment ion spectra (SWATH) and data-dependent acquisition (DDA) modalities, which provided both a comparative analysis and a frequency table that could be used to infer protein abundance in tumor versus cortex. A defining feature of the ccRCC ECM composition characterized in this work is its qualitative similarity to the healthy kidney cortex ECM. Within the detection limits of our analysis, neoplastic transformation does not lead to the de novo expression of matrix molecules, but rather, alters the relative abundance of components. One generalization that emerges from the analysis is that the abundance of BM components is diminished in the ccRCC matrix, while the abundance of IM components increases. Tissue from the healthy kidney cortex can be divided into functional units, which are nephrons and blood vessels, and the interstitial space between them. Nephrons and blood vessels are surrounded by BM, which insulates them from the IM scaffold in which renal fibroblasts are embedded (Figure 2). Considering the breakdown of normal epithelial structure that is characteristic of ccRCC, the abundance of IM and loss of BM seen in tumors is not surprising.

In contrast to healthy cortex, ccRCC ECM is enriched in the IM components collagen VI, fibronectin, tenascin C, fibrin, TGFBI and periostin. Several studies indicate that these matrix components can influence the behavior of tumor cells, suggesting that the ECM environment could be an important determinant of tumor aggressiveness. Collagen VI is abundantly expressed in tumors from several organs, including breast [55], colon, and lung [56]. It promotes the survival of tumor cells [57] and fibroblasts [58] and has been shown to stimulate tumor angiogenesis [59]. The tumor content of collagen VI also promotes the invasive behavior of breast cancer cells [60] and colorectal cancer cells [61]. Studies of xenografted ccRCC cells show that collagen VI expression increases tumor size [62].

In summary, many lines of evidence support a role for collagen VI in promoting the aggressive behavior of tumor cells. Collagen VI is also an important driver of organ fibrosis and has been shown to directly promote the differentiation of cardiac fibroblasts to the hypersecretory myofibroblast phenotype, which is central to scarring after infarction [63]. The phenotypic conversion of vascular mural cells and fibroblasts to myofibroblasts is also central to fibrosis of the kidney [64], and the abundance of collagen VI in the ccRCC tumor may promote the development of a fibrotic local environment through the conversion of interstitial cells to myofibroblasts that secrete additional ECM components, the most characteristic being fibronectin [65]. Because collagen VI is a major component of the healthy kidney IM, it is difficult to argue that its presence in the tumor is sufficient to convert the matrix environment to a pro-fibrotic and pro-tumorigenic state. Rather, the contribution that collagen VI makes in the establishment of these pathologic environments may be contextual. Collagen VI is physically associated with a wide variety of other matrix proteins such as fibronectin [66], decorin [67,68], biglycan [68], fibulin [69], and collagens I [70] and IV [71], which were also identified as abundant components of ccRCC ECM in our analysis [23]. The transition from healthy cortex to tumor tissue with its associated alterations in structure and the abundance of these proteins may generate a higher-order matrix complex combining BM and IM components that are normally sequestered in distinct tissue compartments. This matrix complex is predicted to have unique properties and may influence the behaviors of both tumor and stroma cells very differently from the healthy cortex BM and IM compartments.

Another possibility is that processing of the collagen VI peptide fragment endotrophin (ETP) increases in the tumor environment. ETP is a peptide generated by proteolytic cleavage of the carboxy-terminus of collagen VI α3 [72], normally occurring in the adipocytes of white adipose tissue. As fat mass increases, so does circulating ETP, and the finding that this “matrikine” acts as a driver of malignant tumor growth has led investigators to conclude that it may contribute to the association between obesity and cancer [73,74]. ETP is abundant in mammary tumors, which are surrounded by adipocytes [72]. Little is reported on the mechanism by which adipocytes release ETP from collagen VI α3, but one intriguing observation is that this mechanism is engaged when 3T3-L1 fibroblasts are chemically induced to accumulate lipid and form adipocytes [72]. It is possible that the ETP cleavage mechanism could also be activated in renal epithelial cells as they undergo the metabolic transformation to ccRCC tumor cells, in which fatty acid oxidation is repressed and lipids accumulate in the cytoplasm [75]. Alternately, ETP may be locally supplied by perirenal fat, the abundance of which has been correlated with cancer progression in ccRCC [76]. The increased relative risk of kidney cancer in obese individuals [77] strongly suggests a role for adiposity in ccRCC, and further investigation is required to understand whether ETP circulates to or is produced locally in ccRCC tumors.

Fibronectin can be found circulating as a soluble dimer, and in tissue as an insoluble ECM component [78]. Although fibronectin proteins are encoded by a single gene, functionally distinct isoforms are generated by alternate splicing. Circulating fibronectin produced by hepatocytes is largely devoid of the EIIIA and EIIIB domains present in the insoluble protein that is deposited in the tissue [79]. As previously described, fibronectin establishes fibrillar networks through integrin interactions, and these may be disturbed in VHL-inactivated ccRCC cells [35]. The understanding that fibronectin interacts with multiple other matrix molecules has led to a model of fibronectin as an organizer of higher-order matrix structure [80]. Together with its interacting partner collagen VI [66], it would be predicted to function as an adhesive for the majority of the most abundant matrix proteins identified in ccRCC tumors. Binding partners of particular relevance to the ccRCC matrisome include fibrin [81], periostin [82] and tenascin C [83]. Studies of cultured ccRCC cells in which fibronectin was knocked down suggest a role for fibronectin in promoting cell growth and migration in ccRCC [84]. The correlation of fibronectin expression with patient survival in the TCGA database shows an inverse relationship, suggesting detrimental effects of fibronectin in tumors [85]. A study of tumor tissue from 270 ccRCC patients that scored fibronectin protein expression in the membrane, cytoplasm and nucleus of tumor cells found higher disease-related mortality in patients with cytoplasmic fibronectin [86]. The genetic status of VHL was not defined in this patient cohort, and the impact of the loss of VHL-dependent fibronectin fibrillogenesis was not evaluated.

Tenascin C closely resembles fibronectin, shares receptor-binding properties and is expressed in the stroma of many solid tumors [87]. Little has been reported on the role of tenascin C in ccRCC, but a study of prognostic significance based on the correlation of clinical outcomes with histopathological evaluation of tumors from 137 patients showed that patients with tenascin C-positive tumors had a significantly lower survival rate and suggested that they also had increased risk of metastasis [88]. Studies in a model of glioblastoma have shown that tenascin C promotes proliferation and reduces cell adhesion by reducing the binding of fibronectin to its receptor syndecan 4, suggesting a cell biological mechanism for clinical correlation in ccRCC [89].

TGFBI is a TGFβ-induced protein that is secreted into the extracellular space [90] where it can bind to collagen VI [91] and fibronectin [92]. It shares almost 50% similarity with periostin and has been assigned roles as both a tumor suppressor and a tumor promoter in different experimental systems [93]. Studies in cultured ccRCC cells indicate that TGFBI promotes migration and invasion [50], suggesting a tumor-promoting role. A fluorescent in situ hybridization study of genes with predicted copy number variations in ccRCC confirmed copy number gain in TGFB1 and concluded that it may have a tumor-promoting role [94].

Periostin plays an important role in the organization of collagens by physically interacting with the enzymatic complex that covalently cross-links collagens and enhancing its activity [95]. It binds to both fibronectin and tenascin C [82], which are abundantly represented in the ccRCC ECM [23]. A study of tumor material from 1007 RCC patients concluded that elevated amounts of periostin in tumor cells were correlated with sarcomatoid differentiation and more aggressive tumor cell behavior [96].

Proteoglycans such as HSPG2 (perlecan), lumican and biglycan were identified in tumor ECM, although their abundance was similar to that in healthy cortex [23]. Proteoglycans are composed of a protein core to which glycosaminoglycan chains are covalently bound. HSPG2 is represented in the stroma of several tumor types [97], where it is predicted to bind tenascin C and modify growth factor signaling, including VEGF and FGF, by increasing the binding of these ligands to their receptors [98,99]. Lumican and biglycan are small leucine-rich proteoglycans (SLRPs) that promote collagen fibrillogenesis [100,101]. No reports correlating the properties of ccRCC tumors or patient outcomes with HSPG2 or biglycan could be found for this review. A study of 128 ccRCC patients, including 14 with matched primary ccRCC tumors and pulmonary metastases, correlated lumican expression with metastasis-free and overall patient survival [102].

Many of the studies reviewed in this section suggest a connection between the abundance of specific ECM components, tumor aggressiveness and patient outcome. However, none of the ECM components reviewed in this section are unique to ccRCC tumors, and all are represented in healthy kidney cortex where there is abundant representation of healthy tubule epithelial cells. One major difference between tumor tissue and the neighboring healthy cortex is that epithelial cells in the healthy cortex are insulated from the IM (Figure 2). In the tumor, the BM structure is broken down and these cells are exposed directly to the components of the IM. This may provide novel matrix signals that influence the process of transformation and tumor growth.

## 4. Cellular Sources of ECM in ccRCC Tumors

Single-cell transcriptome analysis of the healthy adult proximal tubule, the presumed cell of origin for ccRCC, reveals little or no expression of genes that encode essential BM components such as collagen IV, laminin or nidogen. Other tubule epithelial cells of the cortex such as the collecting duct, as well as fibroblasts, do express these components [103]. From this analysis, it appears likely that the role of the proximal tubule cell may primarily be to organize the BM on which it sits. Components of the IM such as collagens V and VI are expressed by the fibroblast in the healthy kidney. Whether the proximal tubule cell ever becomes a significant source of ECM from the point of its initial transformation to the time of tumor diagnosis decades later, or whether the matrix composition of the evolving tumor is defined by other cell types, are interesting questions. Histopathology shows that endothelial cells, pericytes, fibroblasts, and immune and inflammatory cells accumulate in and around the tumor, and the matrix proteins secreted by these cells may largely or entirely define ECM composition as the tumor grows. Deconvolving the sources of specific ECM components is challenging, but single-cell transcriptome analysis of ccRCC does provide an insight into the expression of matrix genes. Single-cell transcriptome analysis of human tumors has been reported [104], and reanalysis of this data, focusing on matrix components, indicates that tumor fibroblasts are a major ECM source [23]. This finding agrees with observations of tumors in other organ systems [105], where ECM from myofibroblasts is a major determinant of the tumor microenvironment. Interestingly, fibroblasts that were identified in the Young et al. single-cell dataset [104] were segregated into two clusters; the smaller subset had high smooth muscle actin (ACTA2) expression, suggesting they are myofibroblasts. This subset expressed many of the ECM components identified by mass spectrometry of tumor material: collagen VI, fibronectin, lumican, laminin and collagen XII. Based on this, we hypothesize that a minor subset of fibroblasts in an activated myofibroblast state plays a major role in generating ECM in ccRCC (Figure 3). A sparse population of ACTA2-expressing cells can be seen surrounding clusters of clear cells in ccRCC tissue via immunostaining [23]. The location of these cells indicates that they deposit the matrix within the stromal network around tumor cells that contains vessels and immune cells. These ccRCC myofibroblasts also express the gene encoding the angiotensinogen protease renin, which activates the renin–angiotensin system (RAS) to promote vasoconstriction. Renin-expressing cells have previously been reported in ccRCC stroma [106], and they have been proposed as cancer stem cells [107]. Single-cell analysis indicates that they are instead a myofibroblast subpopulation of ccRCC CAFs. Renin is expressed in a subpopulation of cells in the vessel walls of the healthy kidney [108], which maintains blood pressure by secreting renin in response to neural and chemical cues [109]. Involvement of the RAS system in ccRCC progression is suggested by the clinical observation that patients treated with angiotensin inhibitors display improved survival in metastatic ccRCC [110]. While the interpretation of this finding is complex because of the systemic effects of angiotensin inhibition, recent laboratory-based studies have shown that RAS inhibition prevents ccRCC tumor colony formation, indicating that the pathway does act directly on tumor tissue [111]. The local myofibroblast activation of RAS in ccRCC is, therefore, an intriguing possibility.

## 5. ECM as Therapeutic Target

In addition to the effects of specific ECM components on tumor cells, reviewed above, ECM determines the rigidity of tumor tissue. Measurements of ccRCC tumor samples have shown that the rigidity of the tumor tissue is reduced compared with the surrounding cortex, and that the collagen content of ccRCCs does not correlate with rigidity [112]. A study with a sample size large enough to correlate tumor rigidity with tumor progression and patient outcomes in ccRCC has not yet been reported, but based on studies of other tumor types [113], one would expect increases in tumor matrix rigidity due to collagen crosslinking to promote tumor progression and the invasive behaviors of ccRCC tumor cells. As described in Section 2, VHL is thought to play a major role in controlling the fibrillogenesis of the matrix, and its loss in ccRCC would be predicted to weaken the matrix. Counteracting this, the gene encoding lysyl oxidase (*LOX*) is under the control of HIF transcription factors, which are transcriptionally active when VHL is lost. LOX crosslinks collagens, and in vitro studies of ccRCC cells have shown that LOX increases matrix rigidity [114]. Controlling matrix rigidity may be therapeutically tractable through the modification of LOX, an approach that has been clinically tested using the antibody Simtuzumab, which targets LOXL2. While this is an attractive concept, no clinical improvement was seen in randomized clinical trials of patients with colorectal adenocarcinoma treated with FOLFIRI [115] or in those with pancreatic adenocarcinoma treated with gemcitabine [116] when Simtuzumab was added to the treatment regimen. While Simtuzumab is specific to LOXL2, there are multiple LOX homologs, and it is possible that targeting these proteins more broadly may be required to see an effect in the tissue.

The CAF is a major source of tumor matrix, and numerous approaches have been developed to target this cell type. While the cellular origins of CAFs remain unclear, and presumably vary between cancer types, there is agreement on the common signaling pathways that promote their activation [24]. Inflammatory signaling, matrix composition and matrix rigidity are contributing factors, as is the expression of the fibroblast activation protein (FAP) and signaling through the TGFβ and hedgehog (Hh) pathways. Therapeutic targeting of CAFs is an attractive strategy because these cells are generally non-transformed and prospects for the evolution of escape mechanisms are presumed to be minimal in contrast to the transformed tumor cells. Strategies have been developed to specifically target fibroblast activation pathways including FAP, TGFβ and Hh. A summary of clinical trials using drugs that target each of these pathways is reviewed in [24]. The CAF has multiple functions in addition to the production of the matrix, including growth factor and cytokine production, which establish a local microenvironment, potentially with distinct subcompartments within the tumor. Because of their diverse functions, it is not clear if CAFs promote tumor growth and aggressiveness in all contexts. For example, the depletion of CAFs in pancreatic cancer reduced survival in a mouse model [117]. Too little is known about CAFs in ccRCC to make predictions regarding the benefit of targeting them therapeutically. Further investigation of their role in cell signaling, immunomodulation and matrix production is needed to accurately define the functions of these cells in ccRCC. Recent progress in the genetic modeling of ccRCC in mice [118] and the development of 3D culture systems for studies of ccRCC tumor cell interactions with CAFs [23] now provide powerful platforms for the study of these cells.

## 6. Conclusions

The ECM environment controls many fundamental properties of tumors, including their proliferation, vascularization and invasion, and is therefore a key determinant of their malignancy. The tumor ECM is a complex mix of components, characteristic of the organ from which the tumor arises. Each component has distinct physicochemical and signaling properties, and in aggregate, the ECM defines the overall layout and rigidity of the tumor tissue. In ccRCC, the relationship between the ECM and VHL is of particular interest since VHL is lost in the majority of these tumors. In addition to its role in controlling the availability of HIF transcription factors, which is foundational for cellular oxygen sensing, VHL regulates integrins, controls the organization of matrix proteins and suppresses the expression of matrix-degrading enzymes. The loss of VHL promotes a loosely organized and angiogenic matrix environment that may represent one of the first steps in tumor formation. Later in the evolution of the tumor, multiple cell types accumulate, including CAFs, which express matrix proteins and are predicted to be a major determinant of ECM composition in ccRCC at the time of diagnosis. While this tumor ECM composition qualitatively resembles that of the healthy neighboring kidney cortex in which the tumor arises, there are important differences. One is the quantitative difference between the healthy cortex ECM and the tumor ECM; tumor ECM contains a higher proportion of interstitial matrix components and a lower proportion of basement membrane components. Another important difference is the breakdown of tissue compartments in the tumor with mixing of ECM components that are physically separated in the healthy kidney cortex. Numerous studies reviewed in this work suggest effects of specific ECM components on growth and invasive behaviors of ccRCCs. Studies using novel animal and in vitro models will refine our understanding of the roles of individual ECM components in promoting or inhibiting tumor growth and aggressiveness, and will also answer questions regarding how tissue rigidity is determined in ccRCC.

## Figures and Tables

**Figure 1 cancers-14-04072-f001:**
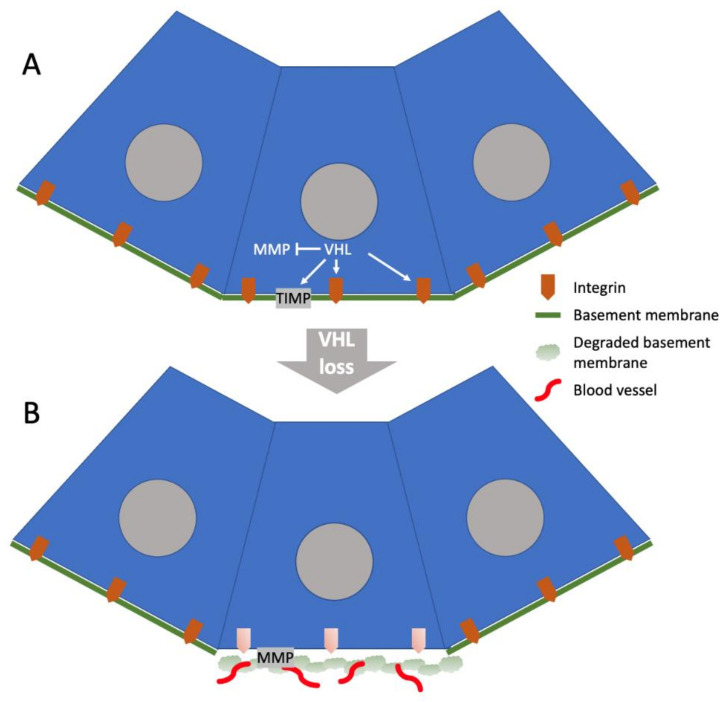
Hypothetical model for changes in the ECM environment of the proximal tubule following loss of VHL. (**A**) VHL controls the half-life of integrins that apply force to fibronectin molecules associated with the basement membrane, which regulates their fibrillogenesis. VHL also suppresses expression of matrix metallopeptidases (MMPs) and upregulates tissue inhibitors of metalloproteinases (TIMPs), ensuring integrity of the basement membrane. (**B**) Following inactivation of *VHL*, fibronectin fibrillogenesis is reduced and MMPs become active, degrading the basement membrane and promoting angiogenesis.

**Figure 2 cancers-14-04072-f002:**
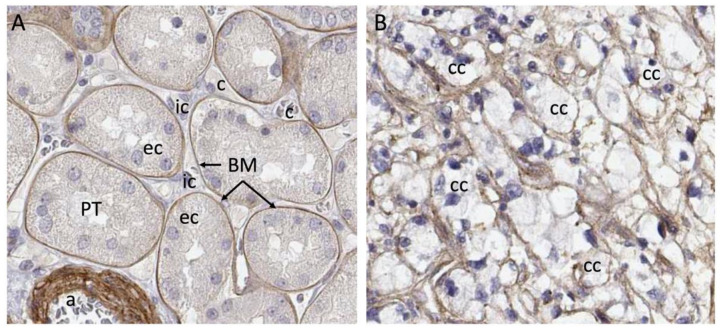
Breakdown of compartments in ccRCC. (**A**) Laminin a5 immunostaining shows strict basement membrane segregation of proximal tubule epithelial cells from the interstitium. (**B**) Cancer cells in ccRCC, which are thought to originate from the proximal tubule cell are embedded in matrix containing laminin a5, but there is no clear segregation of the interstitial cell space from the epithelium. Abbreviations: a—arteriole; BM—basement membrane; c—capillary; cc—clear cell; ec—epithelial cell; ic—interstitial cell. Images from Human Protein Atlas www.proteinatlas.org, licensed under the Creative Commons Attribution-ShareAlike 3.0 International License. https://www.proteinatlas.org/ENSG00000130702-LAMA5/tissue/kidney#img and https://www.proteinatlas.org/ENSG00000130702-LAMA5/pathology/renal+cancer#img were accessed on 12 August 2022.

**Figure 3 cancers-14-04072-f003:**
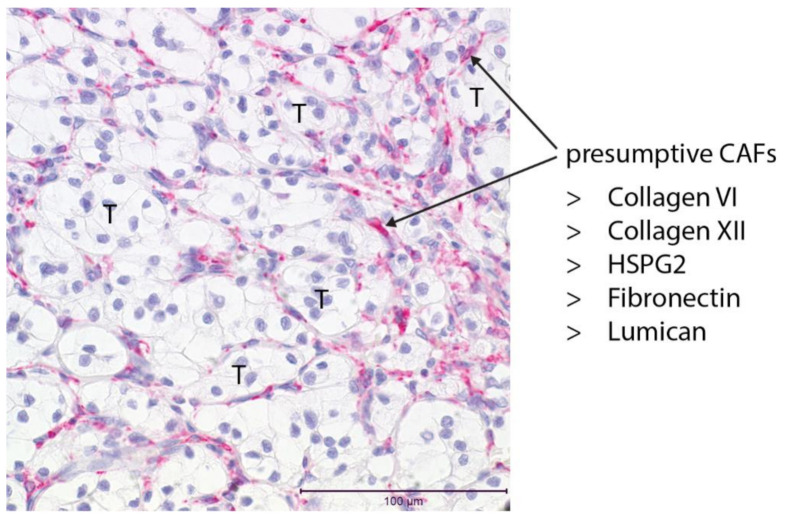
The putative CAF population in ccRCC. Smooth muscle actin immunostaining (red) with hematoxylin nuclear counterstain (blue) reveals a sparse population of cells in the stroma surrounding clusters of tumor cells (T). Single-cell transcriptome analysis indicates that this cell population actively expresses many of the components of the tumor ECM, including collagen VI, collagen XII, HSPG2, fibronectin and lumican.

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
