# Peer review of "The Extracellular Matrix Environment of Clear Cell Renal Cell Carcinoma"

_cancers, 2022, doi:10.3390/cancers14174072_

Round 1
Author Response
Many thanks to the reviewer for their positive assessment of the manuscript.
Reviewer 2 Report
1. It will be better to have more figures to present your key concepts. ANKITA ANIRBAN: Timeliness and figures make a huge difference (Nature 2020: How to write a superb literature review)
2. Many forms of proteoglycans are present in virtually all extracellular matrices. Proteoglycan should be discussed here.
3. Suggest to add the references: Impact of Extracellular Matrix Components to Renal Cell Carcinoma Behavior. Majo S, et al. Front Oncol. 2020; Extracellular matrix structure. Theocharis AD, et al. Adv Drug Deliv Rev. 2016; The matrix in cancer. Cox TR. Nat Rev Cancer. 2021.
4. There are many sentences without its in-text citations, such as line 91,99, 120, 155, ---, and so on.
5. A redundant “period” appears in line 136. A redundant “that” appears in line 164.
6. The format of integrin receptors “aVb3” in line 113 is different from the others.
Author Response
Many thanks to the reviewer for their thorough evaluation of the manuscript. The following changes have been made in response to the reviewer’s comments. Substantially changed sections in the text are in red:
- Add more figures. A third figure has been added to visualize the CAF population within ccRCC.
- Discuss proteoglycans. A section has been added starting at line 310.
- Add citations. These have been added to lines 54, 60, and 280.
- Many sentences do not contain in-text citations. The reviewer points out lines 91, 99, 120. These are introductory statements at the beginnings of paragraphs and the in-text citations supporting them are provided in the body of the paragraph.
- Redundant text in lines 136 and 164. Removed.
- Integrin aV differs from other integrin nomenclature eg a5. This is a feature of the integrin naming scheme. aV (ITGAV) is CD51 while a5 (ITGA5) is CD49.